# Evaluation of Comparative Field Studies for Root and Onion Harvester with Variable Angle Conveyor

Alexey Dorokhov, Alexander Aksenov, Alexey Sibirev *, Maxim Mosyakov, Nikolay Sazonov
and Maria Godyaeva

FSBSI "Federal Scientific Agronomic and Engineering Center VIM", Moscow 109428, Russia
* Correspondence: sibirev2011@yandex.ru; Tel.: +7-964-584-3518

**Abstract:** The long-term storage of vegetable crops and potatoes in storage, even with a small content of soil and plant impurities in the heap, does not ensure the preservation of the required quality indicators of marketable products. The lack of modern technological foundations for improving the design parameters of machines for harvesting vegetables and potatoes leads to the impossibility of eliminating the loss of root crops, their damage, as well as the high-quality process of cleaning from impurities. This circumstance is due to the fact that modern technologies and technical means of harvesting root crops and onions are not able to provide high-quality marketable products with minimal labor costs. In this regard, a modern separating system of the harvesting machine has been developed, which ensures the variation of technological parameters under changing conditions for the harvesting of root crops and onions. The experimental studies of the developed harvester with an experimental separating system made it possible to ensure the high-quality harvesting of potato and onion tubers with the following parameter values: a completeness of separation of more than 98% and a damage to products up to 1.7%, at a speed of movement of 1.7 m/s for the separating system; a completeness of separation of more than 98% and a product damage of up to 1.1% at a speed of up to 1.0 m/s for the harvester; and a separation completeness of more than 98% and a product damage of up to 1.4% at a commercial product extraction depth of 0.02 m. The results of comparative studies on the quality indicators of the machine for harvesting root crops in the harvesting of potato and onion tubers are presented, indicating the prevailing values of the quality indicators of work in the harvesting of potatoes, depending on the change in the regime indicators of the quality of work.

**Keywords:** technological process; rod elevator; lifting angle; displacement; experiment; cleaning machine

## 1. Introduction

The development of agricultural engineering in the Russian Federation cannot be ensured without the development of machines with digital intelligent systems [1–6].

The quality of work is determined by the optimal solution to the problem of providing the required parameters of the machines, both for sowing, harvesting, and the post-harvest processing of crops [7–9], while maintaining the full potential of seeds and grown products [10–12]. The determining indicator of the quality of crop harvesting is their preservation and the minimum amount of losses and damage to marketable products, with an effective value of cleaning from various impurities, which cannot be achieved without the introduction of modern harvesting machines. The main difference between the harvesting of root crops and onions from the harvesting of other agricultural crops is the increased volume of separated soil and plant impurities, therefore, the separating system of machines for harvesting vegetables and potatoes should have a high-throughput [13,14]. However, the separating system of harvesters, represented by bar elevators, does not allow the full cleaning of commercial products from mechanical impurities, and to eliminate this drawback, separation intensifiers of various types have been introduced into their design.

R. Farhadi et. at [15] intensified the process of the separation of potato tubers, the scheme and the general view of which are shown in Figures 1 and 2.

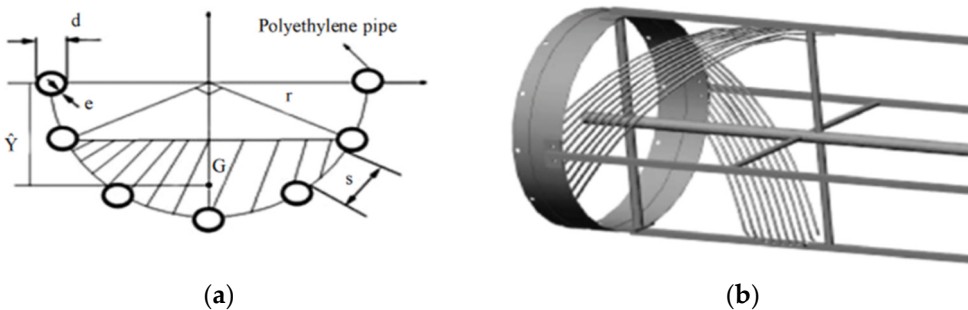

**Figure 1.** Gradation pipes cross-section at a semicircle of the helix ((**a**) scheme; and (**b**) general form): d—the bar diameter; e—bar thickness; s—bar spacing; r—drum radius; G—center of gravity of the drum; and $\hat{Y}$—distance from the center of gravity to the center of the bar.

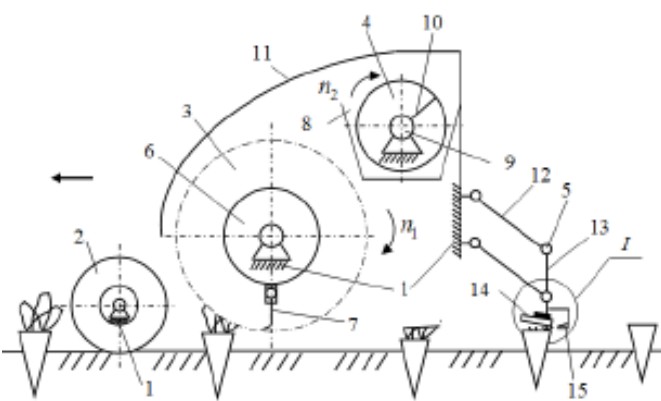

**Figure 2.** Constructive scheme of the improved machine for haulm cutting: 1—chassis frame; 2—support (reference) wheel; 3—rotor; 4—auger; 5—cutter of heads root crops; 6, 9—drum; 7—knife; 8—chute; 10—spiral; 11—casing; 12—parallelogram hinged suspension; 13—bracket; 14—copier; 15—knife; ←—forward movement.

Figure 1 shows the cross-section of the pipes at a semicircle of the helix. For the determination of the pipe diameter, the following points should be attended to: the semicircle radius and perimeter are 15 cm and πr. The initial distance between the pipes is two cm and the final distance is eight cm. The pipes' numbers of helix semicircle generators are 14 at first, for suitable grading. Anything lower than six pipes does not appear at the last pitch. The pipes' size should exist in industrial productions. This equation should be valid for avoiding semicircle defectiveness.

The various stages of piece construction were done with consideration to the presented matters before parts. Figure 2 shows the total side view of the system. Just as observed in the figure, the helix was rotated by a tractor with use of the multipurpose joint, belt and pulley, gearbox, chain, and sprocket wheel. This power transmission system was applied successfully to the reduction of the speed up to a necessary level, and diminished the rotational speed of 540 on 9 RPM. The main disadvantage of the well-known design of the separating device is the turbulent movement of the tuberous heap, which ensures the distribution of the marketable products of tubers along the diameter of the working body, without the presence of a soil layer, which increases the damage to the potato. A known separating device was developed by Ping Yuan Xiong, Xuan Lin, and Yi Wang [16]. A known machine for harvesting table roots was developed by Storozhuk I.M. and Pankiv V.R., the scheme of which is shown in Figure 2, providing an increase in the quality of the cleaning of commercial products as a result of improving the root adapter [17].

However, the disadvantage of this technical solution is the uneven height of the cut of the tops due to the changes in soil density when the machine moves across the field, which leads to a decrease in the product quality as a result of an additional mechanical impact on the root crop and the impossibility of the further utilization of plant impurities remaining on the surface of the root crop.

Thus, the improvement of the design of the digging working bodies of harvesting machines does not fully allow for the improvement of the quality of the harvesting, therefore, it is necessary to carry out the research and development of the functioning elements of the harvesting machine to clean the commercial products from plant and mechanical impurities.

The purpose of the study is to develop separating devices for harvesters that improve the quality of the separation and reduce the damage to commercial products, which is achieved through the determination of the quality indicators of harvesting with an experimental machine for harvesting potatoes and onions.

## 2. Materials and Methods

### 2.1. Bar Elevator with Adjustable Web Angle

The developed bar elevator (RF patent No. 2679734) helps to improve the quality of harvesting, in terms of indicators such as: the damage and completeness of the separation when changing the angle of inclination $\alpha_1$ of the separating system web, depending on the variation in harvesting conditions [13].

In order to change the vertical position of the bar elevator (1), the weight sensors (2) installed on the harvesting machine (3) are used. (Figure 3).

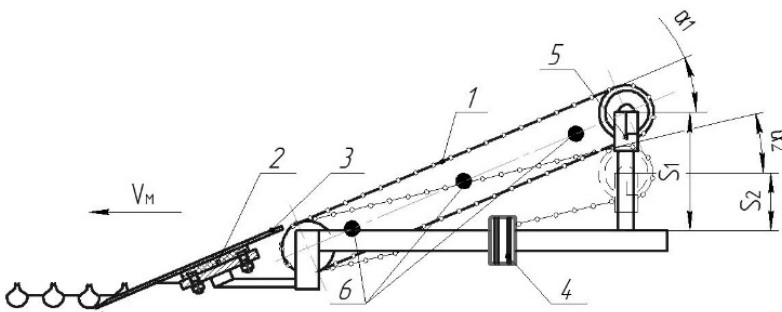

**Figure 3.** Scheme for changing the angle of inclination of the bar elevator conveyor: 1—bar elevator; 2—weight sensor of the digging share; 3—digging share; 4—microcontroller; 5—electric cylinder; and 6—inertial sensor; ←—forward movement.

The mass control device (2) of the heap entering the separation transmits a control signal to the microcontroller (4), and as a result, the web inclination angle is corrected when the mass of the incoming products changes.

The drive for changing the inclination angle of the web of the bar elevator (1), consisting of electric cylinders (5), ensures the movement of the rod to the required distance when the heap mass changes and, accordingly, the inclination angle of the elevator web (1) changes.

The control of the soil that has passed through the openings of the elevator is provided by inertia sensors (6), located along the length of the elevator (1).

In spite of this, the amount of sieved soil will be characterized by peak amplitude values. Thus, the evaluation of the separation efficiency of the elevator is carried out according to the amplitude values.

If the sieving of the soil on the rod elevator is below the required value set by the microcontroller (4), the angle $\alpha_1$ of the inclination of the web of the rod elevator (1) is corrected in the set range of values.

Having determined the mass of a heap of root crops and onions on the digging share, the controller with a time delay T, with the movement of the bar elevator, transmits a control signal to move the rod of the electric cylinders.

Field studies on the separating rod elevator with an adjustable tilt angle of the machine for harvesting root crops and onions (Figure 3) were carried out on the fields of the farm "Tsirulev E.P." This was in the Privolzhsky district of the Samara region in 2019 for the harvest of Meduza onion, and also in 2022 for the harvest of the Red Scarlett potato variety at Krasnaya Gorka LLC, in the Kolyshleysky district of the Penza region. The studies were carried out in accordance with STO AIST 8.7-2013 "Machines for harvesting vegetables and melons. Methods for assessing functional indicators".

The scheme of the experiment included the performance of comparative studies on the quality indicators of onion and potato harvesting by a harvester with a developed separating system on a planting area of 0.04 ha.

Before performing studies on the experimental sample of the separating systems, the physical and mechanical properties of the soil were determined, and the quality indicators of the separation of a heap of turnips and potato tubers were also determined. Other factors considered were the soil of the digging plowshare, the forward speed $v_K$ of the movement of the machine for the harvesting of the root crops and onions, as well as the forward speed $v_{EL}$ of the movement of the rod elevator web on the indicators of the quality of harvesting. The assessment of the quality of the technological process of cleaning was carried out according to the following indicators:

- The damage D to the bulbs/tubers;
- The completeness of the separation $\nu$ of a heap of onions/tubers [17,18].

The extraction of the marketable products from the soil was carried out at a depth $h_L$ of digging from 0.02 to 0.06 m, with a value of its change equal to 0.1 m.

The step of changing the speed $v_K$ of the harvester varied within 0.2 m/s in the range from 1.0 to 1.8 m/s.

By changing the gear ratio of the separating system, the speed $v_{EL}$ of the rod elevator varied from 1.0 to 1.8 m/s, with a step of 0.2 m/s.

The quality indicators of the machine with an experimental separating system were determined after passing the accounting plot, and the sifted products on the tarpaulin along the entire length of the harvested area were collected.

In the selected heap, its fractional composition was determined: bulbs/tubers, free soil, and the soil associated with marketable products.

Before performing studies on the experimental sample of the separating systems, the physical and mechanical properties of the soil were determined, and the quality indicators of the separation of a heap of turnips and potato tubers were also determined. For conducting research on the site, the following were insured: that there was medium loamy chernozem, that the field relief was even, that the field contour was close to rectangular, and that the length of the rut was 350 m. Other factors considered were the soil of the digging plowshare, the forward speed $v_K$ of the movement of the machine for the harvesting of the root crops and onions, as well as the forward speed $v_{EL}$ of the movement of the rod elevator web on the indicators of the quality of harvesting. The assessment of the quality of the technological process of cleaning was carried out according to the following indicators:

### 2.2. Bar Elevator with Asymmetric Shakers

Due to the fact that the technology of growing potatoes differs from onion crops in-depth, their patterns of planting, tillage, and, accordingly, the depth of digging during harvesting also differ. This is due to the fact that when onion ripens, more than 1/2 of the bulb is located on the soil surface, which contributes to the ripening, the formation of a good shirt, and facilitates mechanized harvesting while reducing the flow of soil impurities to the separating working bodies, which is a distinctive feature of potato harvesting, since the tuber nest is located at a digging depth of 15–20 cm and, together with marketable products, a large amount of the soil provides impurities.

Therefore, to intensify the process of separating the potato tubers from the soil impurities, separation intensifiers with an asymmetric arrangement of shakers are installed in the separating system of the harvester (Figure 4).

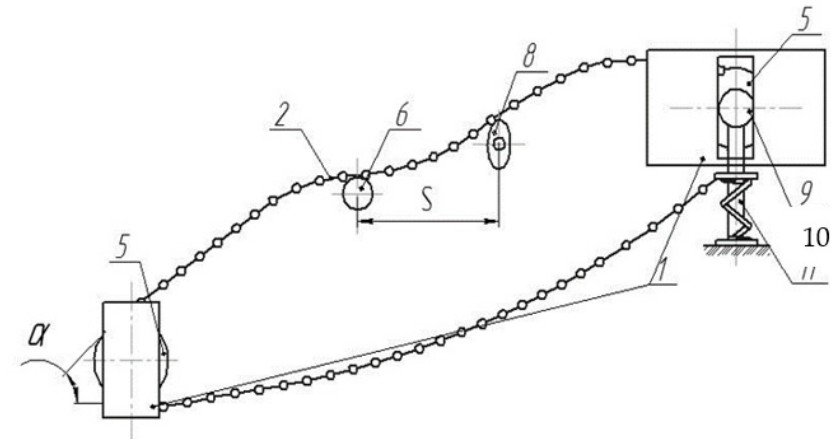

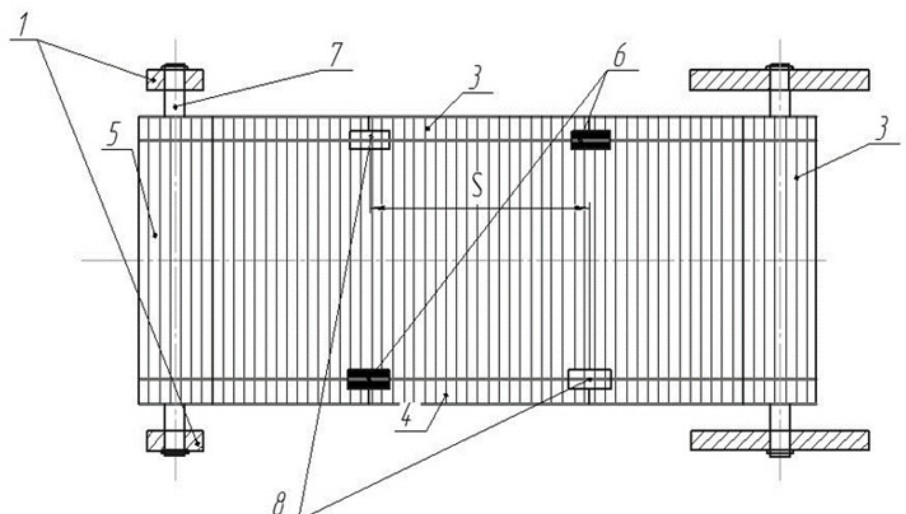

**Figure 4.** Scheme of separating rod elevator: 1—frame; 2—rod elevator; 3, 4—branches of rod elevator; 5—leading roller; 6—support roller; 7—driven roller; 8—elliptical shaker; 9—driven roller shaft; and 10—the mechanism for regulating vertical movement.

The separating rod elevator with asymmetrically installed passive elliptical shakers (RF patent No. 2638190) and with an adjustable angle of the rod elevator provides a reduction in the damage to and an increase in the quality of the separated products. This is as a result of reducing the effect of the vertical component of the gravity of the roots and bulbs to a minimum, as well as increasing the uniformity of the distribution of a pile of root crops and bulbs along the separating surface, when the angle $\alpha$ of the inclination of the rod elevator changes as a result of the changes in soil and climatic conditions for harvesting root crops, onions, and potatoes [17].

The separating conveyor of the machine for harvesting root crops, onions, and potatoes (Figure 4) contains a bar elevator (2) mounted on frame (1), undersides (3), and branches of a rod elevator (4), of which a leading roller (5) is installed, and support (6) and driven (7) rollers are mounted on a frame (1).

The results of field studies on the developed machine for harvesting potatoes of the Red Scarlett variety obtained dependencies that determined the performance indicators for the completeness of the cleaning and damage to tubers in the conditions of Krasnaya Gorka LLC in 2022.

The studies were carried out after the desiccation of the tops of a potato plant, with its subsequent removal on medium sandy soils at a moisture content along the entire length of the accounting plot, within 18–22%.

When conducting research within the conditions of the above farms to determine the quality indicators of harvesting, with a duration of harvesting an area of 0.1 ha, the bulbs and tubers were sorted into intact and damaged.

The damaged bulbs/tubers included products with slight and severe damage caused by machine harvesting.

Light damage to the bulbs/tubers includes:

- bulbs/tubers, bare up to 1/2 with cracks 1 mm deep, up to 10 mm long.

Severe product damage includes:

- bulbs/tubers more than 1/2 bare, with cracks more than 1 mm deep and more than 10 mm long, with dents more than 10 mm.

The damage to the bulbs/tubers was determined by the formula:

$$P = \frac{G_P}{G_{CT} - G_P} \cdot 100\%, \tag{1}$$

where

$G_P$—the weight of the damaged standard bulbs/tubers in a heap, kg;
$G_{CT}$—the mass of the separated bulbs/tubers in a heap, kg.

The completeness of the separation of a heap of onions/tubers was determined by the formula:

$$\nu = \frac{\nu_P^I - \nu_P^K}{\nu_P^I} \cdot 100\%, \tag{2}$$

where

$\nu_P^I$—the mass of the soil impurities in the initial heap, kg;
$\nu_P^K$—the mass of the soil impurities in the container (non-isolated impurities), kg.

The quality of the harvester was determined as follows.

At the beginning of the accounting plot, with the non-stop movement of the harvesting unit, on a signal, a tarpaulin was placed under the separating rod elevator, into which the entire harvested mass was collected.

In the process of passing the plot, a tarpaulin was unwound behind the machine, onto which a heap fell after the separation.

Next, samples were taken from the surface of the tarpaulin from the entire territory of the accounting plot. At the same time, the fractional composition of the heap was determined, which took into account: the bulbs/tubers, the free soil, and the soil associated with the bulbs/tubers.

### 2.3. Research Processing Methods

The reliability of the obtained data is ensured by the methods of mathematical processing and statistical analysis used for the research results and multivariate analysis, which included the use of licensed mathematical software packages for PC: "Microsoft Excel", "STATISTICA-10.0", "Math CAD 2020".

At the same time, the concepts and elements generally accepted in variation statistics that characterize the variation series were used: the average variation—X, the standard deviation—σ, and the coefficient of variation—ν. Each of the main elements was determined according to the known formulas of variation statistics.

This made it possible to determine the accuracy of the experimental data and establish the acceptable limits within which they are sufficiently reliable.

To determine the number of intervals (K) for the varying values of the parameters of the size–mass characteristics of the tubers, we will use the empirical relationship:

$$K = \sqrt{n}, \tag{3}$$

where n = number of tubers, pcs; and $K = \sqrt{100} = 10$.

The sampling range:

$$R = x_{max} - x_{min}, \tag{4}$$

where $x_{max}$ and $x_{min}$ are the maximum and minimum values of the investigated feature.

The interval of the investigated feature:

$$D = R/K. \tag{5}$$

### 3. Results and Discussion

Based on the results of the experimental data processing, graphs were plotted for the dependence of the completeness of separation of the marketable products of bulbs $v_B$ and tubers $v_T$, as well as their damage ($D_b$ and $D_t$), on the mode and technological parameters of the harvesting machine ($h_L$ and $v_P$) and the separating rod elevator ($v_{EL}$) with an adjustable angle canvas tilt. The research results are presented in Figures 5–9.

With a change in the depth of digging onions and potato tubers, there is a variation in the completeness of the separation of and damage to marketable products, which is explained by the presence of an additional soil layer between the roots and the working surface of the bar cloth, with an increase in the depth of the digging while reducing the damage and increasing the completeness of the separation, regardless of the harvested crop (Figure 7A,B)

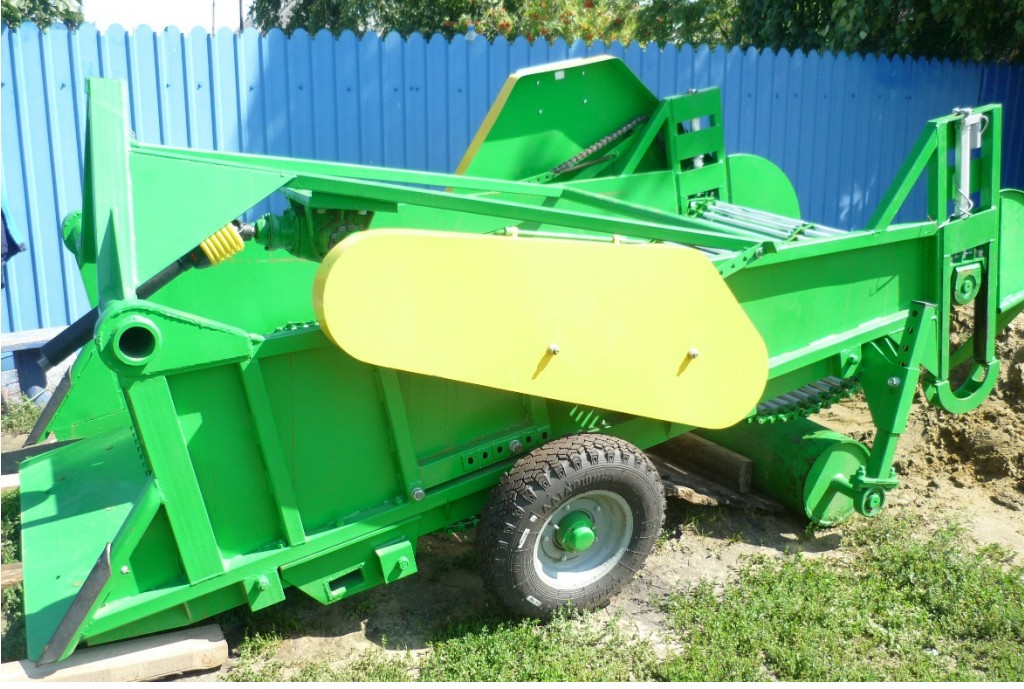

**Figure 5.** General view of the harvesting machine for root crops and onions, equipped with a separating rod elevator with an adjustable blade inclination angle and a receiving plowshare for digging/picking up root crops and bulbs.

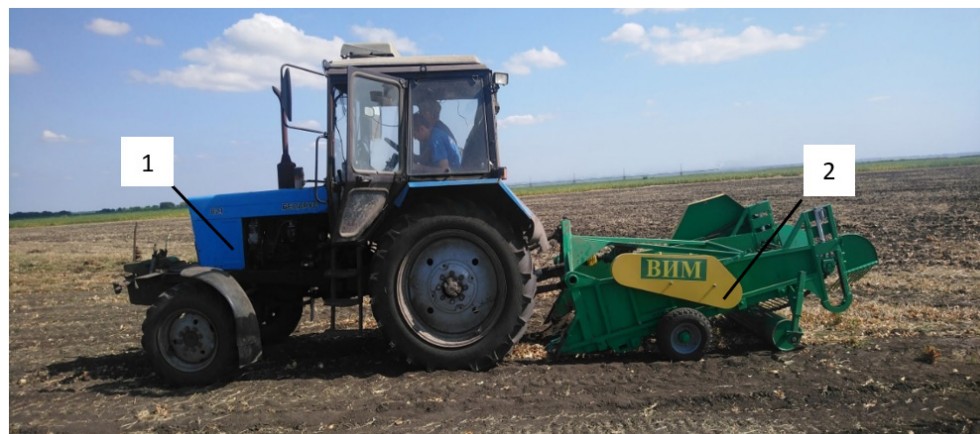

**Figure 6.** General view of the harvesting unit for harvesting root crops and onions, equipped with a separating rod elevator with an adjustable blade inclination angle: 1—MTZ 1221 tractor; and 2—machine for harvesting root crops and onions.

The greatest damage to 1.8% and 1.4% of the commercial products of onions and potatoes, respectively, is observed with a decrease in the depth of digging to 4 cm, which is explained by the difference in their physical and mechanical properties, and the formation of a more stable peel in the potato tubers than a scaly shirt in bulbs.

The difference between the minimum 1.1% and maximum 1.9% damage to the root crops within the identical values of the studied technological parameter is no more than 0.4%, and is expressed by the correlation dependence:

$$\begin{cases} D_B(h_L) = 1.88 - 0.47{\cdot}h_L - 0.07{\cdot}h_L^2, \\ D_T(h_L) = 1.39 - 0.85{\cdot}h_L + 0.53{\cdot}h_L^2. \end{cases} \tag{6}$$

An obvious circumstance is an increase in the completeness of the cleaning of the root crops (Figure 7B), with a decrease in the depth of digging, which is due to a decrease in the supply of soil impurities to the cleaning devices and a simultaneous increase in damage to the marketable products.

The maximum completeness of the cleaning of the bulbs was more than 96%, and is observed at a depth of digging of up to 4 cm, which, when extracting potato tubers from the soil, provides 95.4% of the completeness of the separation.

The minimum value of the completeness of the separation of the root and tuber crops was 91%, and is observed at a depth of 10 cm of digging in the potato tuber nest (which is 91.6% in comparison with the commercial onion products), however, with an increase in the depth of digging to 12 cm, an increase in the completeness of the cleaning is observed by 0.4% and is explained by an increase in the time of the separated mass on the working surface of the rod elevator, in view of increasing its load and reducing the forward speed of its movement. Differences between the completeness of the cleaning of the studied root crops with a change in the depth of digging are observed, on average, up to 0.5%, and are expressed by a correlation dependence:

$$\begin{cases} v_B(h_L) = 98.6 - 2.66{\cdot}h_L + 0.3{\cdot}h_L^2, \\ v_T(h_L) = 98.4 - 3.56{\cdot}h_L + 0.45{\cdot}h_L^2. \end{cases} \tag{7}$$

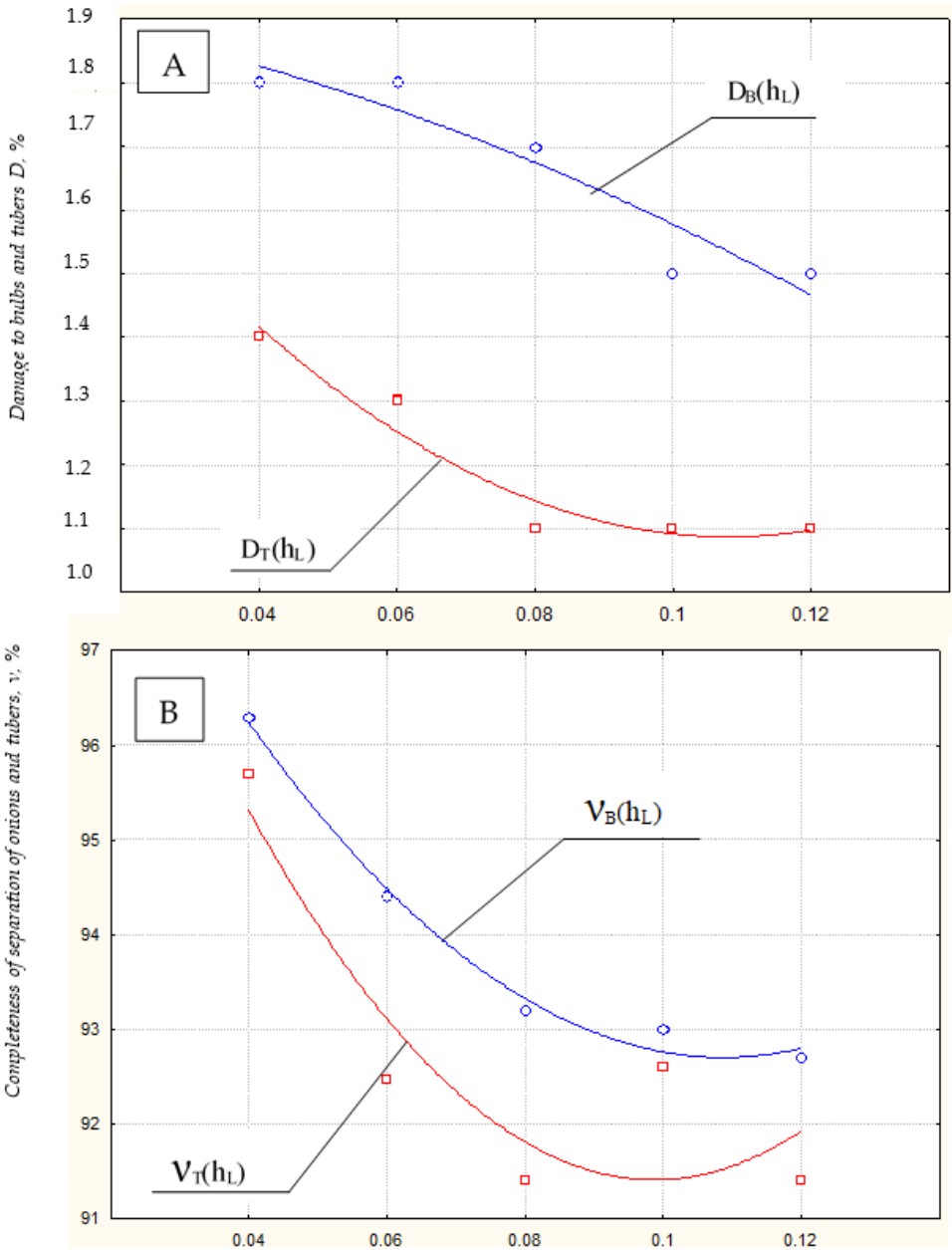

**Figure 7.** Dependence of: (**A**) damage to bulbs $D_B$ and tubers $D_T$ %; and (**B**) completeness of separation of onions $v_B$ and tubers $v_T$% of the separating rod elevator with an adjustable blade inclination angle from the depth $h_L$ of digging share into the soil.

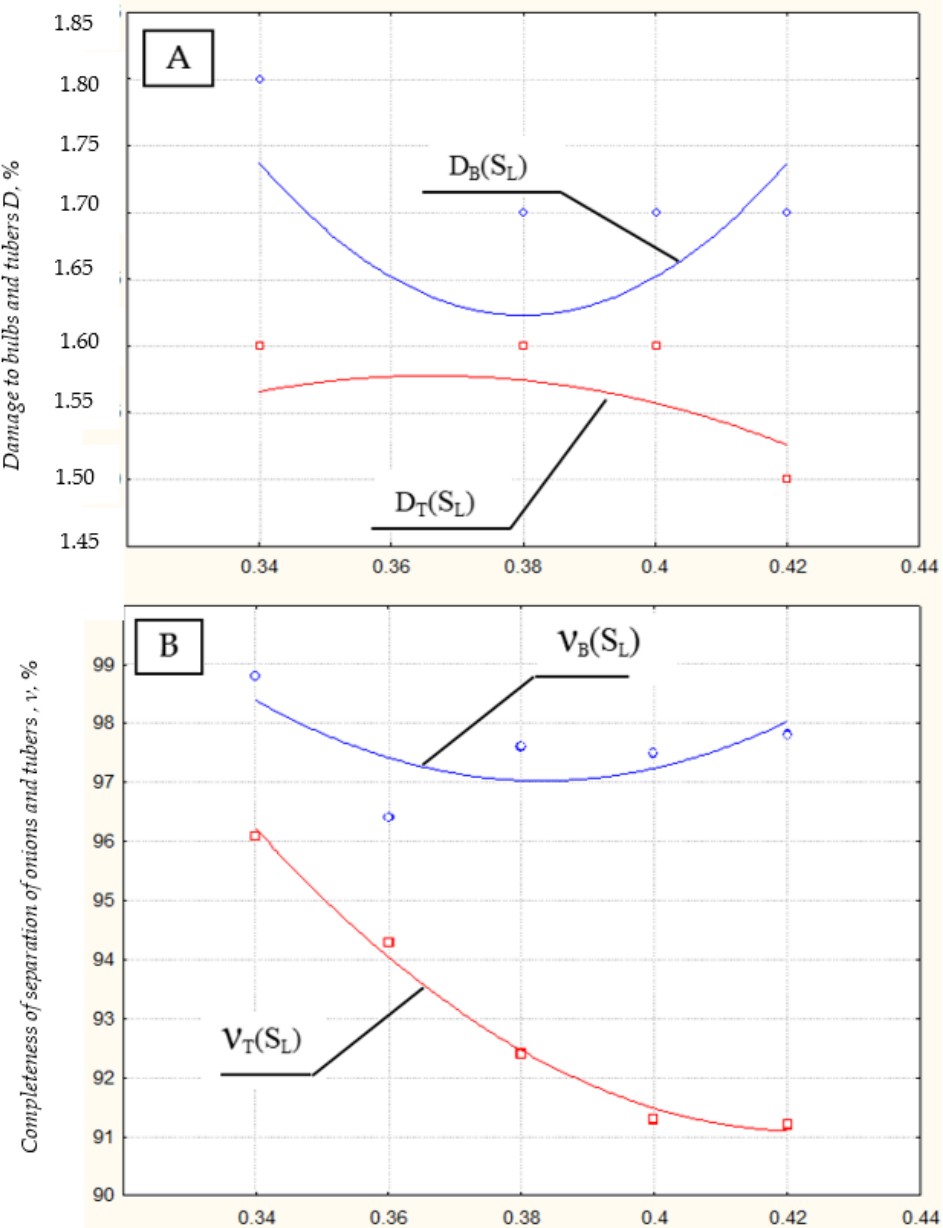

**Figure 8.** Dependence of: (**A**) damage to bulbs $D_B$ and tubers $D_T$, %; and (**B**) completeness of separation of onions $v_B$ and tubers $v_T$% of the separating rod elevator with an adjustable blade inclination angle from the center distance $S_L$ between the shakers.

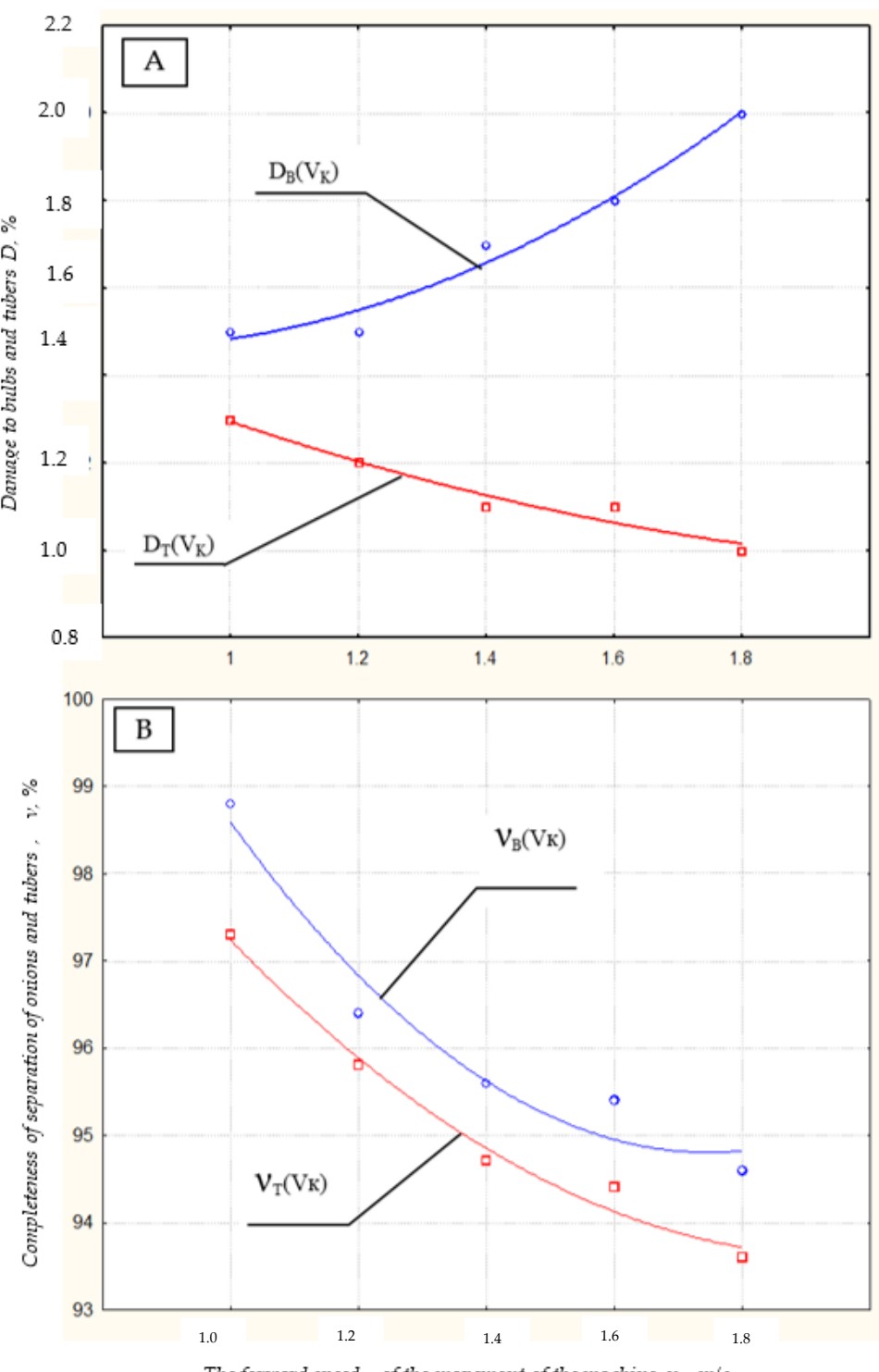

**Figure 9.** Dependence of: (**A**) damage to bulbs $D_L$ and tubers $D_K$%; and (**B**) completeness of separation of onions $v_B$ and tubers $v_T$% of the separating rod elevator with an adjustable blade inclination angle from the forward speed $v_K$ of the machine.

An increase in the quality of the harvesting of potato and onion tubers using a separating rod elevator with asymmetrically installed shakers is ensured by increasing the time for the discrete comparison of impurities with the slotted openings of the working body, as a result of moving not only along the length, but also the width of the rod elevator. At the same time, the maximum amount of the damage to the bulbs is no more than 1.8% at a constant digging depth of 0.1 cm and a forward speed of the rod elevator $v_{EL}$ and machine $v_K$ within 1.0 and 1.2 m/s, respectively. The difference between the maximum and minimum damage to the potato tubers in the studied range of the shaker locations is no more than 0.3%, with a minimum value of 1.52%. In addition, the combination of asymmetrically located shakers on the rod elevator provides a better process of potato tuber separation within 96%, which is 2.3% lower compared to its use in onion harvesting at a technological parameter value of 0.34 cm.

With an increase in the center distance to the maximum allowable value of 0.42 m, a decrease of 91.5% in the completeness of separation is observed in the cleaning of the potato tubers, which is lower than the minimum value of the cleaning of the bulbs by 7%, and is explained by a decrease in the transverse amplitude of the oscillation of the web, due to the connectivity of the potato tubers in the tuber nest of the plant.

The performance indicators are expressed by the correlation dependence:

$$\begin{cases} D_B(S_L) = 1.88 - 0.17 \cdot S_L - 0.02 \cdot S_L^2, \\ D_T(S_L) = 1.54 + 0.03 \cdot S_L - 0.07 \cdot S_L^2. \end{cases} \tag{8}$$

$$\begin{cases} v_Л(S_Л) = 99.2 - 2.84 \cdot S_L + 0.43 \cdot S_L^2, \\ v_K(S_Л) = 98.1 - 3.16 \cdot S_L + 0.35 \cdot S_L^2. \end{cases} \tag{9}$$

The graphic dependences of the change in the quality indicators of the harvesting of potatoes and onions using a rod elevator with an adjustable blade inclination angle on the forward speed $v_K$ of the machine, are shown in Figure 9.

An increase in the studied technological indicator of potato harvesting leads to a decrease in the quality indicators of the minimum values of 1.0% of damage to the tubers, and the 93.8% completeness of separation to the maximum of 1.2% for damage and 97.3% of the completeness of separation.

At the same time, it should be noted that there is a distinctive feature in terms of the damage to onion commercial products, with an increase in the forward speed of the machine from 1.5% to 2.0%, which is explained by an increase in the collisions between the bulbs due to the thickening of their planting, and an increase in the supply of commercial products for separation.

The difference between the minimum values of the damage to bulbs and tubers reaches 1.5% and 1.0%, respectively, with maximum values of 1.2% and 2.0% damage, and is expressed by the equations of parabolic functions:

$$\begin{cases} D_B(v_K) = 1.88 - 0.17 \cdot v_K - 0.02 \cdot v_K^2, \\ D_T(v_K) = 1.54 + 0.03 \cdot v_K - 0.07 \cdot v_K^2. \end{cases} \tag{10}$$

The difference between the quality of the cleaning by groups of commercial products is no more than 2%, with an increased completeness of separation of 97.3% for the harvesting of onions and a minimum value for the harvesting of potatoes of 93.8% with a correlation dependence:

$$\begin{cases} v_B(v_K) = 98.7 - 2.56 \cdot v_K + 0.27 \cdot v_K^2, \\ v_T(v_K) = 98.2 - 1.82 \cdot v_K + 0.15 \cdot v_K^2. \end{cases} \tag{11}$$

Upon analyzing the graphical dependencies (Figure 10), we can say that the maximum damage value is observed for the commercial onion products and is 1.9%, which is 1.7% for potato tubers.

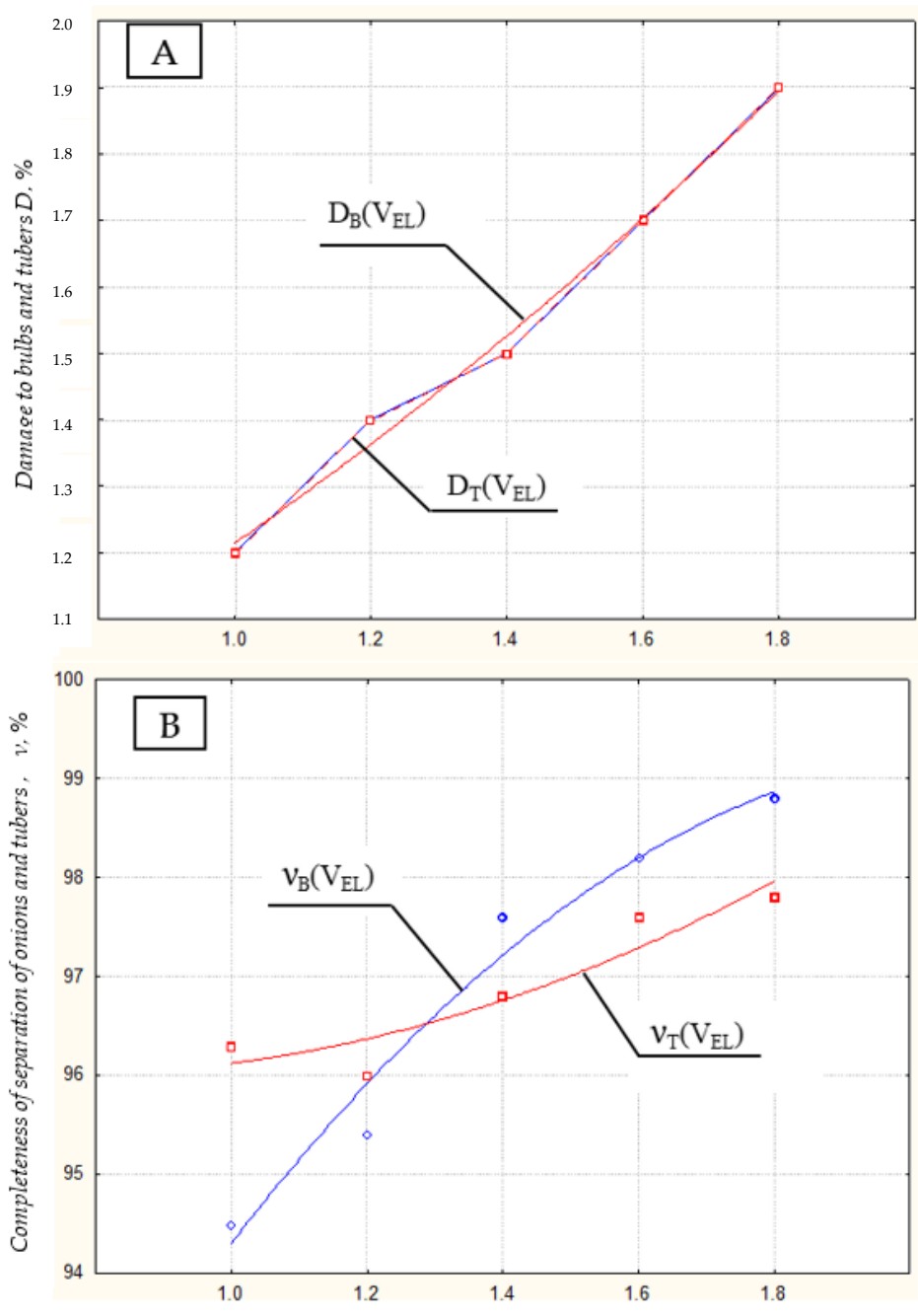

**Figure 10.** Dependence: (**A**) damage to bulbs $D_B$ and tubers $D_T$%; and (**B**) completeness of separation of onions $v_B$ and tubers $v_T$% of the separating rod elevator with an adjustable blade inclination angle from the translational speed $v_{EL}$ of the rod elevator.

The minimum damage value of 1.2% is for all the groups of the commercial products, and increases with an increase in the forward speed of the rod elevator in the range of 1.0 to 1.8 m/s. The correlation between the damage to the marketable products of potato tubers and onion bulbs is determined by the expression:

$$D_B, \ D_T(v_{EL}) = 1.08 - 0.12 \cdot v_{EL} + 0.7 \cdot v_{EL}^2. \tag{12}$$

The maximum separation completeness of more than 98.8% is observed in onion harvesting, which is higher than the completeness of the potato tubers' cleaning by 1%, at a forward speed of the rod elevator of 1.8 m/s.

At the minimum value of the studied indicator of the harvesting machine, equal to 1 m/s, the completeness of the cleaning has a minimum value for potato tubers of 96%, which is 2% higher than that for onion harvesting.

In addition, the optimal value of the forward speed of the rod elevator, at which the approximation of the cleaning completeness curves is 96.8% at a speed value of 1.3 m/s, is described by the correlation dependence:

$$\begin{cases} v_B(v_{EL}) = 92.4 - 2.08 \cdot v_{EL} - 0.15 \cdot v_{EL}^2, \\ v_T(v_{EL}) = 96.2 + 0.31 \cdot v_{EL} + 0.07 \cdot v_{EL}^2. \end{cases} \tag{13}$$

According to the results of the research, it follows that the optimal ratio of the quality indicators of onion harvesting is ensured when crossing curves approximate the completeness of separation at 95.5% and the damage to the bulbs at 1.2% at a forward speed of 1.38 m/s of the machine for the harvesting of root crops and onions.

The reliability of the conducted studies was assessed by the calculated value of the mathematical expectation $M(X)$, and the normal law of the distribution of the damage to the commercial products on a bar elevator is:

$$M(X) = 0.3. \tag{14}$$

Student's distribution quantile:

$$T = qt\left(1 - \frac{\alpha}{2}, \nu\right) = 2.012. \tag{15}$$

Calculation of statistics criterion and distribution of a random variable over intervals submitted in Table 1.

**Table 1.** Calculation of statistics criterion and distribution of a random variable over intervals.

| | Observed | Cumulative | Percent | Cumul. % | Expected | Cumulative | Percent | Cumul. % | Observed |
|---|---|---|---|---|---|---|---|---|---|
| 2.5 | 0 | 0 | 0 | 0 | 0.26 | 0.26 | 1.79 | 1.79 | −0.26 |
| 2.55 | 0.5 | 1 | 6.66 | 6.66 | 0.47 | 0.74 | 3.17 | 4.97 | 0.52 |
| 2.6 | 1 | 3 | 13.33 | 20.0 | 0.98 | 1.73 | 6.56 | 11.54 | 1.01 |
| 2.65 | 1.5 | 3 | 0 | 20.0 | 1.67 | 3.41 | 11.15 | 22.69 | −1.67 |
| 2.7 | 2.5 | 8 | 33.33 | 53.33 | 2.32 | 5.73 | 15.53 | 38.22 | 2.67 |
| 2.75 | 2.5 | 8 | 0 | 53.33 | 2.65 | 8.39 | 17.72 | 55.95 | −2.65 |
| 2.8 | 2.5 | 13 | 33.33 | 86.66 | 2.48 | 10.88 | 16.59 | 72.54 | 2.51 |
| 2.85 | 2 | 0 | 0.00000 | 86.66 | 1.91 | 12.79 | 12.73 | 85.27 | −1.91 |
| 2.9 | 1 | 15 | 13.33 | 100.0 | 1.21 | 13.99 | 8.01 | 93.29 | 0.79 |
| Infinity | 1 | 15 | 0 | 100.0 | 1.0 | 15.0 | 6.7 | 100.0 | −1.0 |

Graphically, the observed and expected frequencies are determined by constructing scatter plots (Figure 11).

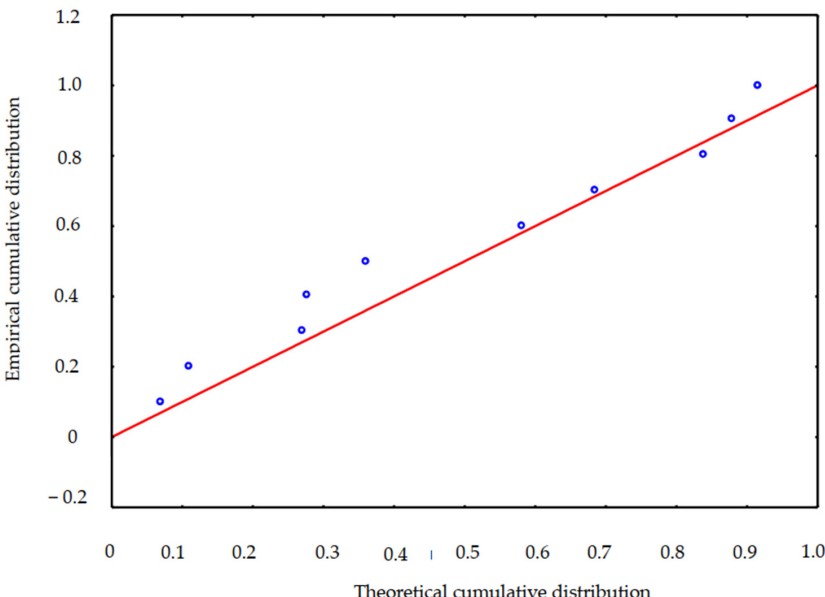

**Figure 11.** Plot of observed values (empirical cumulative distribution) versus expected frequencies (theoretical cumulative distribution).

The mathematical expectation M(X) of the normal distribution law for the separation of the potato tubers and bulbs on the developed cleaning device:

$$M(X) = 0.2. \tag{16}$$

Student's distribution quantile:

$$T = qt \left(1 - \frac{\alpha}{2}, \upsilon\right) = 2.012. \tag{17}$$

Calculation of statistics criterion and distribution of a random variable over intervals submitted in Table 2.

**Table 2.** Calculation of statistics criterion and distribution of a random variable over intervals.

|          | Observed | Cumulative | Percent | Cumul. % | Expected | Cumulative | Percent | Cumul. % | Observed- |
|----------|----------|------------|---------|----------|----------|------------|---------|----------|-----------|
| 95       | 0        | 0          | 0       | 0        | 0.16     | 0.16       | 1.08    | 1.08     | −0.16     |
| 95.5     | 1        | 1          | 6.66    | 6.66     | 0.31     | 0.47       | 2.06    | 3.15     | 0.69      |
| 96       | 1        | 2          | 6.66    | 13.33    | 0.68     | 1.15       | 4.58    | 7.73     | 0.31      |
| 96.5     | 0        | 2          | 0       | 13.33    | 1.26     | 2.42       | 8.43    | 16.16    | −1.26     |
| 97       | 0        | 2          | 0       | 13.33    | 1.93     | 4.35       | 12.88   | 29.04    | −1.93     |
| 97.5     | 2.4      | 8          | 40      | 53.33    | 2.44     | 6.81       | 16.32   | 45.37    | 3.55      |
| 98       | 2.6      | 11         | 20      | 73.33    | 2.57     | 9.38       | 17.15   | 62.53    | 0.42      |
| 98.5     | 0        | 11         | 0       | 73.33    | 2.24     | 11.62      | 14.96   | 77.49    | −2.24     |
| 99       | 3        | 14         | 20      | 93.33    | 1.62     | 13.24      | 10.82   | 88.31    | 1.37      |
| 99.5     | 0        | 14         | 0       | 93.33    | 0.97     | 14.22      | 6.49    | 94.81    | −0.97     |
| 100      | 1        | 15         | 6.66    | 100      | 0.48     | 14.71      | 3.23    | 98.04    | 0.51      |
| Infinity | 0        | 15         | 0       | 100      | 0.29     | 15         | 1.95    | 100      | −0.29     |

Graphically, the observed and expected frequencies are presented in the scatter plot (Figure 12).

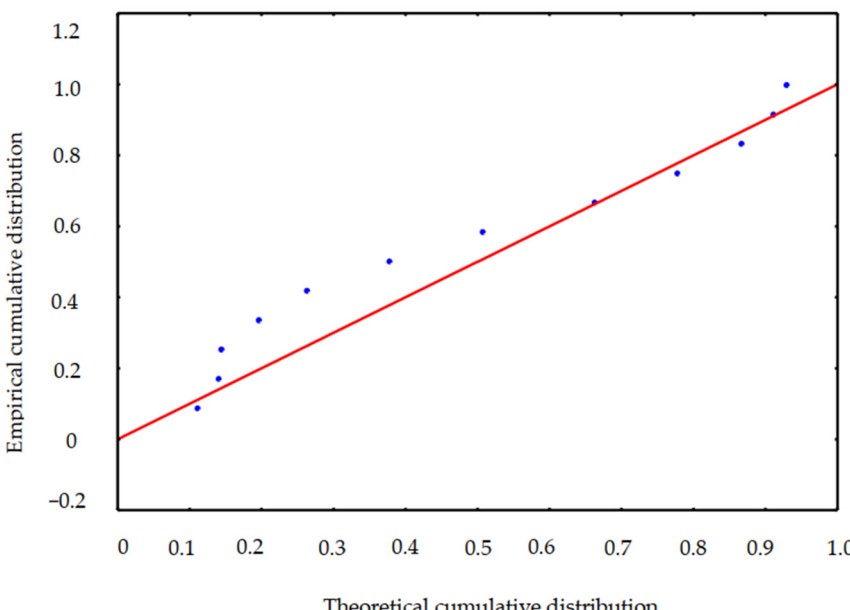

**Figure 12.** Plot of observed values (empirical cumulative distribution) versus expected frequencies (theoretical cumulative distribution).

The statistical processing of the results of the experimental studies, according to the criteria $S_y^2$ и $S_{LF}^2$, are shown in Figure 13.

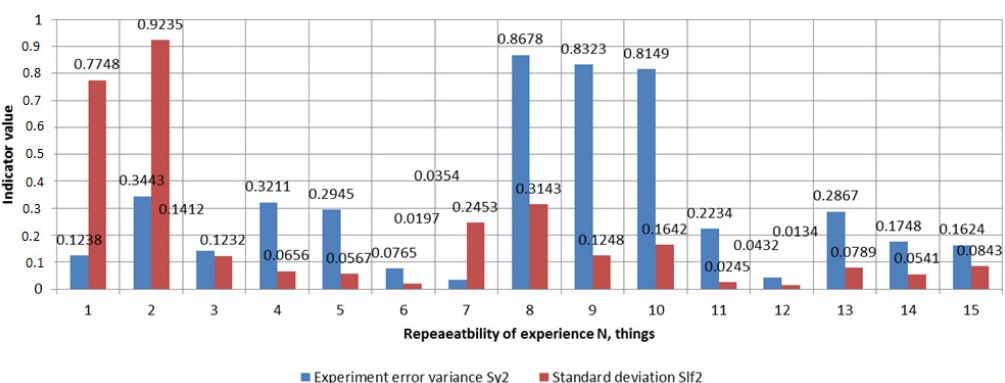

**Figure 13.** Statistical characteristics of experience error.

The results of the conducted field studies on a machine for harvesting root crops and onions, equipped with a bar elevator with an adjustable blade inclination angle and an asymmetric arrangement of shakers, showed the high-quality performance of the technological process of separating turnips and potato tubers at the optimal values of the parameters: a completeness of separation of more than 98%, and damage to the products of up to 1.7% at a speed of movement of 1.7 m/s of the separating system; a completeness of separation of more than 98%, and s product damage of up to 1.1% at a speed of up to 1.0 m/s of the harvester, and a separation completeness of more than 98%, and product damage of up to 1.4% at a commercial product extraction depth of 0.02 m.

The results obtained from the studies are consistent with the previously known data on the development and testing of machines in terms of the cleaning of the marketable products of root and tuber crops.

The obtained research results are consistent with the previously known directions for improving the design of harvesters.

Research performed by V.P. Khambalkar and colleagues developed an onion harvester that has been helpful in harvesting the onion bulb effectively. The design of the onion

harvester greatly depended on the working width, the depth of the operation, and the conveying capacity. The agro-technical data of the crop has been collected from the university field regarding the depth of the onion bulb, the spacing between the plant and the row-to-row distance, the height of crop, and the polar and equator dimensions of the onion bulb. The overall size of the harvester is $1020 \times 600 \times 300$ mm. The width-to-depth ratio for the optimum operation performance is chosen to be 6:1. The depth of the operation according to the agro-technical requirement comes to be 10 cm. The working width of the machine, with respect to the power availability and the crop sowing characteristics, comes to be 60 cm [18].

## 4. Conclusions

For the manufacture and production of the technical means of a potato harvesting machine, it is recommended to choose technological parameters from experimentally confirmed values that ensure the high-quality performance of the harvesting process.

Improving the design and technological schemes of the machines for harvesting potatoes and onions, on the basis of the conducted research, can further be used in the system for monitoring the quality indicators of the machines for the harvesting and post-harvest processing of potatoes and other agricultural root crops, which will provide a new level of synthesis of the complex technical systems of agricultural production.

The proposed technological and technical solutions can serve as a basis for expanding the research on and practical implementation of Russia's transition to the sustainable development of vegetable production.

**Author Contributions:** Conceptualization, A.S.; methodology, A.D. and M.G.; software, N.S. and M.G.; validation, A.A.; investigation, A.S.; resources, M.M.; writing—original draft preparation, M.G.; writing—review and editing, A.S.; project administration, A.S.; funding acquisition, N.S. All authors have read and agreed to the published version of the manuscript.

**Funding:** The research was carried out with the financial support of the Russian Science Foundation of the 2022 contest "Conducting research by scientific groups led by young scientists" of the Presidential Program of research projects implemented by leading scientists, including young scientists No. 22-76-10002.

**Institutional Review Board Statement:** The authors declare that they have no known competing financial interests or personal relationships that could have appeared to influence the work reported in this paper.

**Informed Consent Statement:** The authors declare that they have no known competing financial interests or personal relationships that could have appeared to influence the work reported in this paper.

**Data Availability Statement:** The raw data supporting the conclusions of this article will be made available by the authors, without undue reservation.

**Conflicts of Interest:** The authors declare that they have no known competing financial interests or personal relationships that could have appeared to influence the work reported in this paper.

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
