# Peer review of "Evaluation of Comparative Field Studies for Root and Onion Harvester with Variable Angle Conveyor"

_agriculture, doi:10.3390/agriculture13030572_

Round 1

Reviewer 1 Report (Previous Reviewer 1)

Dear authors, the article 'Evaluation of Comparative Field Studies for Root and Onion Harvester with Variable Angle conveyor' is very interesting, however changes are required before publication. Moreover, information provided are not organized, thus making very difficult to understand its useful findings.

In general, English language need to be improved. The current language and style make the manuscript hard to understand.

Also, figures and plots quality need to be improved. 

Abstract: 

line 25: correct unit m/s 

Introduction:

Figure 1 and Figure 2 could be presented as one figure with 2 subsections (a and b)

line 118: change will with may

line 120: change is with was

Materials and Methods:

Some information is difficult to understand. Please clarify how many trials have been carried out, their duration and what authors measured during these trials. Please, clarify how authors evaluated the damages to tubers and bulbs and the complete separation of onions and tubers at different fields and or tractor conditions.

Results:

Table 1 and table 2 should be changed according to journals format.

In plot figures probably a legend may improve the clarity.

Discussion:

In this section results obtained should be compared with other relevant and recent studies present in literature in order to prove the point of the trial.

Conclusion:

Conclusions are fine.

Author Response

Hello dear reviewer! I send you answers to your comments

Reviewer 2 Report (New Reviewer)

Dear Authors

Comments on the reviewed article:

1) I believe that the introduction should be corrected editorially and Figures 1-3 are not very legible. Especially figure 2 is not very legible and descriptive. I would suggest to remove these figures from the introduction and to describe give more literature items.

2) Subsection 2.3 mentions that a statistical analysis is carried out and Section 3 does not contain the results of this analysis.

3) Section 3 is named: Results and Discussion, which lacks discussion and references to the literature. Not a single literature item is given.

4) In the conclusions, I would suggest that lines 455 - 462 be moved to the section Results 

5) Other comments are included in the manuscript 

Author Response

Hello dear reviewer! I send you answers to your comments

Reviewer 3 Report (New Reviewer)

Paper is looking good, as i have seen some revision made on this version,

so please find the following notes:

1- No need to list all author's names for any related work that you need to present in the article such as (R. Farhadi, N. Sakenian, and P., etc, [15], mention this related work as R. Farhadi et. at and then reference number).

2- Cite the patents that you listed in your article in an appropriate manner.

3- Figure 8. overlaps of some text, adjust the position

4-  Some references belong to the same author (Sibirev, A.V.) { Ref No. 1, 3, 4, 5} and it's not recommended to cite 4 related works from the same author and be sure that its not belong to you.

5- Some of your references are not cited well, such as Reference No.3, you have cited it with URL of Research Gate, and the correct citation of that reference as follow:

Sibirev, A. V., Aksenov, A. G., & Mosyakov, M. A. (2018). Experimental laboratory research of separation intensity of onion set heaps on rod elevator. Journal of Engineering and Applied Sciences13(23), 10086-10091.

However, correct all the references that follow the same manner.

Author Response

Hello dear reviewer! I send you answers to your comments

Round 2

Reviewer 1 Report (Previous Reviewer 1)

Dear authors, 

most of the comments have been addressed.

In my opinion the experimental design (farms location, experimental timing, and so on) is still not clear. I suggest to provide additional information in material and methods section (line 216) to improve clarity. It is difficult to understand where, how and when the experimental devices developed were studied. Autohrs sholud make a clear statement about this.

Morover, images quality still needs to be improved.

Author Response

Hello, dear reviewer! I am sending you a corrected version of the article taking into account your comments.

Reviewer 2 Report (New Reviewer)

Dear Authors

Thank you for incorporating my comments and suggestions into the manuscript. The manuscript can be published. 

Author Response

Hello dear reviewer! Thank you for the detailed study of the corrected version of the article.

This manuscript is a resubmission of an earlier submission. The following is a list of the peer review reports and author responses from that submission.

Round 1

Reviewer 1 Report

Dear authors, the article 'Evaluation of Comparative Field Studies for Root and Onion Harvester with Variable Angle conveyor' is very interesting, however the way as it is written make it difficult to understand. As first I suggest and extensive editing of English language to improve manuscript readibility and clarity. Moreover information provided are not enough for a critical comparison with obtained results and are not organizaed at all, thus making very difficult to understand its useful findings. For instance, authors state the aim of the research (it is not clear if is the aim of that one trial) in results and discussion section misleading the reader through the manuscript. However, in my opinion, with a good work of rephrasing, restyling and reorganizing, the manuscript could be considered suitable for publishing in Agriculture.

Abstract:

abstract sholud be shorten and it does not provide a resume of the trials carried out and their results. Authors in this section should delete all the unnecessary information and highlight some of the major findings. 

Introduction:

the introduction is well done, however I think that authors should make an effort to improve the clarity of the introduction leading the reader to understand the purpose of this study at the end of this section. 

Materials and methods:

the experimental design is not clear. Authors should make an effort and guide the reader through the manuscript. It is not clear how they test the machines (which farms, if they harvested onion of potatoes, how muche did they harvested, and so on). Thus I suggest to divide materials and methods section in different subsection order to improve the clarity.

A subsection should be Experimental design so to clarify the number of trials carried out (only 1 for 2019 and 1 for 2022?), the number of  trials carried out with the separating rod elevator with an adjustable inclination angle (RF patent No. 120 2679734) and with the separating rod elevator with asymmetrically installed passive elliptical shakers 203 (RF patent No. 2638190) and all the specifics concerning the tractor, the levels ov forward spped and so on.

Another subsection should be dedicated to the assessements; authors evaluated the damages to tubers and bulbs and the complete separation of onions and tubers at different fields and or tractor conditions. It is necessary to organize them in a clear way. 

A third subsection should be provided to clarify what type of data analysis were carried out with what software and so on. 

Authors here should provide clear information concerning, machines, modifications and trials carried out. It is necessary to divide this part of the manuscript in different subsections and improve the clarity of the tests fulfilled. It is not clear what authors studyied and hoe they evaluated all the different parameters.

Results and discussion:

results are well written, however the discussion of those findings should be futrther implemented by providing a comparison with some relevant reference in order to prove the point of the trial.

Conclusion:

authors should move the information reported in the conclusion in the discussion part. In this part authors should provide a well constructed conclusion which resume the major findings and eventually provide future developement based on these findings. 

References:

Authors should check the references format and correct it (in the manuscript as well). 

Additional details are provided in the file attached.

Reviewer 2 Report

Dorokhov et al., studied and applied a machine for harvesting root crops and onions, equipped with a bar elevator with an adjustable blade inclination angle and an asymmetric arrangement of shakers. The introduction is exhaustive and well explains the state of the art. Overall, the article is somewhat confused and lacks relevant evidence. The machine could serve as a basis for expanding research and practical implementation of Russia's transition to sustainable development of vegetable production. In my opinion, I suggest making major revisions, as reported in the PDF file attached.  The logic in the article needs to be sorted out and a reliable basis needs to be added.

The specific review comments are as follows:

1. The meaning of the characters in Figure 1 is not explained; the labeling of Figure 3 is not clear; the labeling format needs to be unified in the whole text.

2. The meaning of "PTO" in line 68 and "P4" in line 96 should be added.

3. The discussion in lines 104~108 and 388~390 is suggested to be supplemented with its basis.

4. The "VMe" in equation (1) is not commented, please add it.

5. In "2. Materials and Methods", there are conclusive descriptions in lines 120-126 and 203-210, and it is suggested to revise them.

6. It is suggested to add the reliability of the fitted equations expressed in Eqs. (2) to (9).

7. the symbols in equation (5) are incorrectly written; line 391 "Figure 4" should be revised to "Figure 5".
